# GDPP: Learning Diverse Generations using Determinantal Point Processes

## Abstract

Generative models have proven to be an outstanding tool for representing high-dimensional probability distributions and generating realistic looking images. A fundamental characteristic of generative models is their ability to produce multi-modal outputs. However, while training, they are often susceptible to mode collapse, which means that the model is limited in mapping the input noise to only a few modes of the true data distribution. In this paper, we draw inspiration from Determinantal Point Process (DPP) to devise a generative model that alleviates mode collapse while producing higher quality samples. DPP is an elegant probabilistic measure used to model negative correlations within a subset and hence quantify its diversity. We use DPP kernel to model the diversity in real data as well as in synthetic data. Then, we devise a generation penalty term that encourages the generator to synthesize data with a similar diversity to real data. In contrast to previous state-of-the-art generative models that tend to use additional trainable parameters or complex training paradigms, our method does not change the original training scheme. Embedded in an adversarial training and variational autoencoder, our Generative DPP approach shows a consistent resistance to mode-collapse on a wide-variety of synthetic data and natural image datasets including MNIST, CIFAR10, and CelebA, while outperforming state-of-the-art methods for data-efficiency, convergence-time, and generation quality. Our code will be made publicly available.

## 1 Introduction

Deep generative models have gained enormous research interest in recent years as a powerful framework to learn high dimensional data in an unsupervised fashion. Generative Adversarial Networks (GANs) (Goodfellow et al., 2014) and Variational AutoEncoders (VAEs) are among the most dominant generative approaches. They consist of training two networks: a generator (decoder) and a discriminator (encoder), where the generator attempts to map random noise to *fake* data points that simulate the probability distribution of *real* data. . GANs are typically associated with higher quality images compared to VAEs. Nevertheless, in the process of learning multi-modal complex distributions, both models may converge to a trivial solution where the generator learns to produce few modes exclusively, as referred to by mode collapse problem.

To address this, we propose utilizing Determinantal Point Processes (DPP) to model the diversity within data samples. DPP is a probabilistic model that has been mainly adopted for solving subset selection problems with diversity constraints (Kulesza & Taskar, 2011), such as video and document summarization. However, Sampling from a DPP requires quantifying the diversity of $2^N$ subsets, where N is the size of the ground set. This renders DPP sampling from true data to be computationally inefficient in the generation domain. The key idea of our work is to model the diversity within real and fake data throughout the training process, which does adds an insignificant computational cost. Then, We encourage producing samples of similar diversity distribution to the true-data by back-propagating the DPP metric through the generator. This way, generator explicitly learns to cover more modes of real distribution, and accordingly alleviates mode collapse.

Recent approaches tackled mode-collapse in one of two different ways: (1) improving the learning of the system to reach a better convergence point(e.g. Metz et al. (2017); Arjovsky & Bottou (2017)); or (2) explicitly enforcing the models to capture diverse modes or map back to the true-data distribution

(e.g. Srivastava et al. (2017); Che et al. (2017)). Here we focus on a relaxed version of the former, where we use the same learning paradigm of the standard GANs and only change the objective function. The advantage of such an approach is to avoid adding any extra trainable parameters to the trained system while maintaining the same back-propagation steps as the standard GANs. Thus, our model converges faster to a fair equilibrium point where the generator captures the diversity of the true-data distribution while preserving the quality of generations.

**Contribution**. We introduce a new loss function, that we denote Generative Determinantal Point Processes (*GDPP*) loss. Our loss only assumes an access to a generator $G$, a feature extraction function $\phi(\cdot)$, and sampler from true data distribution $p_d$. The loss encourages the generator to diversify generated samples that match the diversity of real data.

This criterion can be considered as a complement to the original adversarial loss which attempts to learn an indistinguishable distribution from the true-data distribution without being specific to diverse modes. We assess the performance of GDPP on three different synthetic data environments, while also verifying the superiority on three real-world images datasets. We compared our approach with state-of-the-art approaches of more complex architectures and learning paradigms. Experiments show that our method outperforms all competing methods in terms of alleviating mode-collapse and generations quality.

## 2 RELATED WORK

Among the tremendous amount of work that tackles the training challenges of Generative Adversarial Networks (GANs), a few methods stood out as significant contributions towards addressing the problem of mode collapse.

**Methods that map the data back to noise.** (Donahue et al., 2017; Dumoulin et al., 2017) are one of the earliest methods that proposed learning a reconstruction network besides learning the deep generative network. Adding this extra network to the system aims at reversing the action of the generator by mapping from data to noise. Likelihood-free variational inference (LFVI) (Tran et al., 2017), merge this concept with learning implicit densities using hierarchical Bayesian modeling. Ultimately, VEEGAN (Srivastava et al., 2017) used the same concept, but the authors did not base their reconstruction loss on the discriminator. This has the advantage of isolating the generation process from the discriminator's sensitivity to any of the modes. Che et al. (2017) proposed several ways of regularizing the objective of adversarial learning including geometric metric regularizer, mode regularizer, and manifold-diffusion training. Mode regularization specifically has shown a potential into addressing the mode collapse problem and stabilizing the GANs training in general.

**Methods that provide a surrogate objective function.** Chen et al. (2016) on the other hand propose with InfoGAN an information-theoretic extension of GANs that obtains disentangled representation of data by latent-code reconstitution through a penalty term in its objective function. InfoGAN includes an autoencoder over the latent codes; however, it was shown to have stability problems similar to the standard GAN and requires stabilization tricks. Ghosh et al. (2018) base the ModeGAN method on the assumption of the availability of sufficient samples of every mode on the training data. In particular, if a sample from the true data distribution belongs to a particular mode, then the generated fake sample is likely to belong to the same mode. The Unrolled-GAN of Metz et al. (2017) propose a novel objective to update the generator with respect to the unrolled optimization of the discriminator. This allows training to be adjusted between using the optimal discriminator in the generator's objective. It has been shown to improve the generator training process which in turn helps to reduce the mode collapse problem. Generalized LS-GAN of Edraki & Qi (2018) define a pullback operator to map generated samples to the data manifold. BourGAN Xiao et al. (2018), with a similar philosophy, additionally draws samples from a mixture of Gaussians instead of a single Gaussian. There is, however, no specific enforcement to diversify samples. Spectral normalization strategies have been recently proposed in the works of Miyato et al. (2018) and SAGAN (Zhang et al., 2018) to further stabilize the training. We note that these strategies are orthogonal to our contribution and could be implemented in conjunction with ours to further improve the training stability of generator models. Finally, improving the Wasserstein GANs of Arjovsky et al. (2017), WGAN-GP (Gulrajani et al., 2017) introduce a gradient penalization employed in state-of-the-art systems (Karras et al., 2018).

**Methods use multiple generators and discriminators.** One of the popular methods to reduce mode collapse is using multiple generator networks to provide a better coverage of the true data distribution. Liu & Tuzel (2016) propose using two generators with shared parameters to learn the joint distribution of the data. The two generators are trained independently on two domains to ensure a diverse generation. However, sharing the parameters guide both the generators to a similar subspace. Also, Durugkar et al. (2017) propose a similar idea of multiple discriminators that are being an ensemble, which was shown to produce better quality samples. Recently, Ghosh et al. (2018) proposed MAD-GAN which is a multi-agent GAN architecture incorporating multiple generators and one discriminator. Along with distinguishing real from fake samples, the discriminator also learns to identify the generator that generated the fake sample. The learning of such a system implies forcing different generators to learn unique modes, which helps in a better coverage of data modes. DualGAN of Nguyen et al. (2017) improves the diversity within GANs at the additional requirement of training two discriminators. In contrast to these approaches, our DPP-GAN does not require the training of extra networks which provides an easier and faster training as well as being less susceptible to overfitting.

Finally, we also refer to PacGAN Lin et al. (2018) as another approach addressing mode collapse. They do that by modifying the discriminator input with concatenated samples to better sample the diversity within real data. Nevertheless, such approach is subject to memory and computational constraints as a result of the significant increase in batch size.

## 3 DETERMINANTAL POINT PROCESS (DPP)

DPP is a probabilistic measure that was introduced in quantum physics (Macchi, 1975) and has been studied extensively in random matrix theory (Hough et al., 2006). It provides a tractable and efficient means to capture negative correlation with respect to a similarity measure, that in turn can be used to quantify the diversity within a subset. A key characteristic in DPP that the model is agnostic about the order of items as pointed out by Gong et al. (2014), and therefore can be used to model data that is randomly sampled from a certain distribution.

A point process $\mathcal{P}$ on a ground set $\mathcal{V}$ is a probability measure on the power set of $\mathcal{V}$ (i.e., $2^N$), where $N = |\mathcal{V}|$ is the size of the ground set. A point process $\mathcal{P}$ is called determinantal if, given a random subset $Y$ drawn according to $\mathcal{P}$, we have for every $S \subseteq Y$,

$$\mathcal{P}(S \subseteq Y) \propto \det(L_S) \tag{1}$$

for some symmetric similarity kernel $L \in \mathbb{R}^{N \times N}$, where $L_S$ is the similarity kernel of subset $S$. $L$ must be real, positive semidefinite matrix $L \preceq I$ (all the eigenvalues of $L$ is between 0 and 1); since it represents a probabilistic measure and all of its principal minors must be non-negative.

$L$ is often referred to as the marginal kernel because it contains all the information needed to compute the probability of any subset $S$ being selected in $\mathcal{V}$. $L_S$ denotes the sub-matrix of $L$ indexed by $S$, specifically, $L_S \equiv [L_{ij}]; i, j \in S$. Hence, the marginal probability of including one element $e_i$ is $p(e_i \in Y) = L_{ii}$, and two elements $e_i$ and $e_j$ is $L_{ii}L_{jj} - 2L_{ij}^2 = p(e_i \in Y)p(e_j \in Y) - L_{ij}^2$. A large value of $L_{ij}$ reduces the likelihood of both elements to appear together. Kulesza & Taskar (2010) proposed decomposing the kernel $L_S$ as a Gram matrix:

$$\mathcal{P}(S \subseteq Y) \propto \det(\phi(S)^\top \phi(S)) \prod_{e_i \in S} q^2(e_i), \tag{2}$$

Here $q(e_i) \geq 0$ may be seen as a quality score of an item $e_i$ in the ground set $\mathcal{V}$, while $\phi_i \in \mathbb{R}^D; D \leq N$ and $||\phi_i||_2 = 1$ is used as an $\ell_2$ normalized feature vector of an item. In this manner, $\phi_i^\top \phi_j \in [-1, 1]$ is evaluated as a "normalized similarity" between items $e_i$ and $e_j$ of $\mathcal{V}$, and the kernel $L_S$ is guaranteed to be real positive semidefinite matrix.

**Geometric interpretation:** $\det(\phi(S)^\top \phi(S)) = \prod_i \lambda_i$, where $\lambda_i$ is the $i^{th}$ eigen value of the matrix $\phi(S)^\top \phi(S)$. Hence, we may visualize that DPP models diverse representations of data because the determinant of $\phi(S)^\top \phi(S)$ corresponds to the volume in n-D represented by the multiplication of the variances of the data (i.e., the eigen values).

**DPP in literature:** DPP has proven to be a tremendously valuable tool when addressing the problem of diversity enforcing such as document summarization (e.g., Kulesza & Taskar (2011); Hong

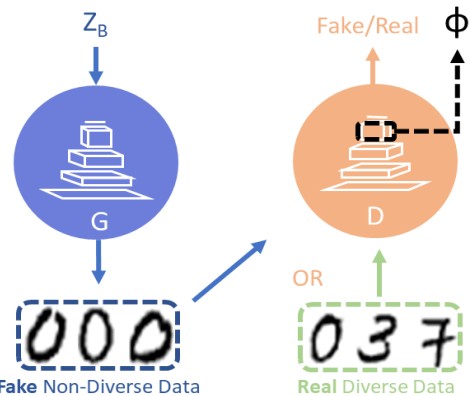

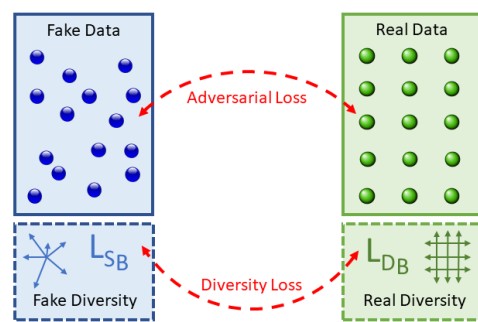

**(a)** Given a generator $G$, and feature extraction function $\phi$, the diversity kernel is constructed as $L_S = \phi^\top \cdot \phi$. We use the last feature map of the discriminator in GAN or the encoder in VAE as the feature representation $\phi$ of a fake/real batch.

**(b)** Using $\phi$ obtained from generated samples, we model their diversity using $L_{S_B}$. We also model the diversity of a real batch by extracting its features and constructing its diversity kernel $L_{D_B}$. Adversarial loss aims at generating similar data points to the real, and diversity loss aims at matching fake data diversity kernel$L_{S_B}$ to real data diversity kernel $L_{D_B}$.

**Figure 1:** We draw inspiration from DPP to model a subset diversity using a kernel. During training, we extract the feature representation of real and fake batches $\phi_{real}$ and $\phi_{fake}$. Then, we construct their diversity kernels: $L_{S_B}, L_{D_B}$. Our loss encourages $G$ to synthesize data of a diversity $L_{S_B}$ similar to the real data diversity $L_{D_B}$.

& Nenkova (2014)), pose estimation (e.g., Gupta (2015)) and video summarization (e.g., Gong et al. (2014); Mahasseni et al. (2017)). For instance, Zhang et al. (2016) proposed to learn the two parameters $q, \phi$ in eq. 2 to quantify the diversity of the kernel $L_S$ using MLPs based on spatio-temporal features of the video to perform summarization. Recently, Hsiao & Grauman (2018) proposed to use DPP to automatically create capsule wardrobes, i.e. assemble a minimal set of items that provide maximal mix-and-match outfits given an inventory of candidate garments.

## 4 GENERATIVE DETERMINANTAL POINT PROCESSES

As illustrated in Fig. 1b, our GDPP loss encourages the generator to sample fake data of diversity similar to real data diversity. The key challenge is to model the diversity within real data and fake data. We discussed in Sec. 3 how DPP is used to model the negative correlation within a discrete data distribution, which is commonly employed as a measure of diversity. Thus, we construct a DPP kernel for both the real data and the generated samples at every iteration of the training process as shown in Fig. 1a. Then, we encourage the network to generate samples that have a similar diversity kernel to that of the training data. To simplify the optimization process, we choose to match the eigenvalues and eigenvectors of the fake data DPP kernel with their corresponding of the real data DPP kernel. Eigenvalues and vectors capture the manifold structure of both real and fake data, and hence renders the matching problem simpler.

During training, a generative model $G$ produces a batch of samples $S_B = \{e_1, e_2, \cdots e_B\}$, where $B$ is the batch size. Our aim is to produce $S_B$ that is probabilistically sampled following the DPP which satisfies:

$$\mathcal{P}(S_B \subseteq Y) \propto \det(L_{S_B}), \tag{3}$$

such that $Y$ is a random variable representing a subset drawn with a generative point process $\mathcal{P}$, and $L_{S_B}$ is the kernel matrix of the subset indexed by $S$, as detailed in Sec. 3. Connecting DPP to the data generation, we assume that $G$ is the point process sampler that generates subset $S_B$ according to $\mathcal{P}$. Let $\phi(S_B) \in \mathbb{R}^{d \times B}$ be a feature representation of the generated subset $S_B$, where $\phi(\cdot)$ is a feature extraction function. Therefore, the DPP kernel is constructed as follows:

$$L_{S_B} = \phi(S_B)^\top \phi(S_B); \quad S_B = G(z_B), \tag{4}$$

where $z_B \in R^{d_z \times B}$ is noise vector inputted to the generator $G$ resulting in the generated subset $S_B$. Let us denote $\phi(S_B)$ a feature representation of a generated batch and $\phi(D_B)$ a feature representation of a true batch. Our aim is to match $\mathcal{P}(S_B \subseteq Y) \propto [\det(L_{S_B}) = \prod_i \lambda^i_{fake}]$ to

$\mathcal{P}(D_B \sim p_d) \propto [\det(L_{D_B}) = \prod_i \lambda_{real}^i]$, where $\lambda_{real}^i$ and $\lambda_{fake}^i$ are the $i^{th}$ eigenvalues of $L_{D_B}$ and $L_{S_B}$ respectively. Thus, our problem is reduced to learn a fake diversity kernel $L_{S_B}$ close to the real diversity kernel $L_{D_B}$. We choose to match those kernels using their major characteristics: eigenvalues and eigenvectors.

Our GDPP loss is composed of two components: diversity magnitude loss $\mathcal{L}_m$, and diversity structure loss $\mathcal{L}_s$ as follows:

$$\mathcal{L}_g^{DPP} = \mathcal{L}_m + \mathcal{L}_s = \sum_i \|\lambda_{real}^i - \lambda_{fake}^i\|_2 - \sum_i \hat{\lambda}_{real}^i \cos(v_{real}^i, v_{fake}^i) \tag{5}$$

where $v_{fake}^i$ and $v_{real}^i$ are the $i^{th}$ eigenvectors of $L_{D_B}$ and $L_{S_B}$ respectively. $\hat{\lambda}_{real}^i$ are the min-max normalized version of its corresponding eigenvalues $\lambda_{real}^i$. We note that $\lambda \geq 0$ for both fake and real data, since both $L_{S_B}$ and $L_{D_B}$ are guaranteed to be positive semidefinite matrices.

Scaling the structure loss aims to induce noise invariance within the eigenvectors similarity learning. This can be seen as alleviating the effect of outlier structures that intrinsically exist within the real data on learning the diversity kernel of fake data. We note that all the eigenvalues of $L_{S_B}$ and $L_{S_D}$ will be real non-negative since both of the kernels are symmetric semi-positive definite. Therefore, the kernels represent a probability measure since none of the principal minors will be negative.

**Integrating GDPP loss with GANs.** For a primary benchmark, we integrate our GDPP loss with GANs. Since our aim is to avoid adding any extra trainable parameters, we utilize features extracted by the discriminator. We choose to use the hidden activations before the last layer as our feature extraction function $\phi(.)$. We apply $\ell_2$ normalization on the obtained features that guarantees constructing a positive semi-definite matrix according to eq. 2. We finally integrate $\mathcal{L}_g^{DPP}$ into the GAN objective by only modifying the generator loss of the standard adversarial loss (Goodfellow et al., 2014) as follows:

$$\mathcal{L}_g = \mathbb{E}_{z \sim p_z}[\log(1 - D(G(z)))] + \mathcal{L}_g^{DPP}. \tag{6}$$

**Integrating GDPP loss with VAEs.** A key property of our loss is its generality to any generative model. Beside incorporating our loss in GANs, we prove it can be also embedded within Variational Auto-Encoders (VAEs) proposed in Kingma & Welling (2013). We use the decoder network as our generator $G$ and the final hidden activations within the encoder network as our feature extraction function $\phi(.)$. To compute $\mathcal{L}^{DPP}$ at the training time, we feed an input training batch $D_B$ to the encoder constructing $L_{D_B}$. We also feed a random Gaussian noise to the decoder that generates a fake batch $S_B$, which we then feed to the encoder to construct $L_{S_B}$. Finally, we compute $\mathcal{L}^{DPP}$ as stated in eq. 2 using the $\ell_2$ normalized features, then add it to the original VAE loss at the training time as follows:

$$\mathcal{L}_{VAE} = \mathbb{E}_{z \sim p(z|x)}[\log\{p(x|z)\}] + KL[p(z|x)||p(z)] + \mathcal{L}^{DPP}. \tag{7}$$

## 5 EXPERIMENTS

In our experiments, we target evaluating the generation based on two criteria: mode collapse and generated samples quality. Due to the intractability of log-likelihood estimation, this problem tends to be non-trivial in real data. Therefore, we start by analyzing our method on synthetic data where we can accurately evaluate the performance. Then, we demonstrate the effectiveness of our method on real data using standard evaluation metrics. We use the same architecture and data on all the competing methods (See appendix A for details).

### 5.1 SYNTHETIC DATA EXPERIMENTS

Mode collapse and the quality of generations can be explicitly evaluated on synthetic data since the true distribution is well-defined. In this section, we evaluate the performance of the methods on mixtures of Gaussian of known mode locations and distribution (See appendix B for details). We use the same architecture for all the models, which is the same one used by Metz et al. (2017) and Srivastava et al. (2017). We note that the first four rows in Table 1 are obtained from Srivastava et al. (2017), since we are using the same architecture and training paradigm. Fig. 2 illustrates the

| | 2D Ring | | 2D Grid | | 1200D Synthetic | |
|---|---|---|---|---|---|---|
| | Modes (Max 8) | % High Quality Samples | Modes (Max 25) | % High Quality Samples | Modes (Max 10) | % High Quality Samples |
| GAN (Goodfellow et al., 2014) | 1 | 99.3 | 3.3 | 0.5 | 1.6 | 2.0 |
| ALI (Dumoulin et al., 2017) | 2.8 | 0.13 | 15.8 | 1.6 | 3 | 5.4 |
| Unrolled GAN (Metz et al., 2017) | 7.6 | 35.6 | 23.6 | 16.0 | 0 | 0.0 |
| VEE-GAN (Srivastava et al., 2017) | **8.0** | 52.9 | 24.6 | 40.0 | 5.5 | 28.3 |
| WGAN-GP (Gulrajani et al., 2017) | 6.8 | 59.6 | 24.2 | 28.7 | 6.4 | 29.5 |
| GDPP-GAN | **8.0** | **71.7** | **24.8** | **68.5** | **7.4** | **48.3** |

**Table 1:** Sample quality and degree of mode collapse on mixtures of Gaussians. GDPP-GAN consistently captures the highest number of modes and produces better samples.

effect of each method on the 2D Ring and Grid data. As shown by the vanilla-GAN in the 2D Ring example (Fig. 2a), it can generate the highest quality samples however it only captures a single mode. On the other extreme, the WGAN-GP on the 2D grid (Fig. 2k) captures almost all modes in the true distribution, but this is only because it generates highly scattered samples that do not precisely depict the true distribution. GDPP-GAN (Fig. 2f,l) creates a precise representation of the true data distribution reflecting that the method learned an accurate structure manifold.

**Performance Evaluation:** At every iteration, we sample fake points from the generator and real points from the given distribution. Mode collapse is quantified by the number of real modes recovered in fake data, and the generation quality is quantified by the % of High-Quality Samples. A generated sample is counted as high-quality if it was sampled within three standard deviations in case of 2D Ring or Grid, and ten standard deviations in case of the 1200D data. We train all models for 25K iterations, except for VEEGAN which is trained for 100K iterations to properly converge. At inference time, we generate 2500 samples from each of the trained models and measure both metrics. We report the numbers averaged over five runs with different random initialization in Table 1. GDPP-GAN clearly outperforms all other methods, for instance on the most challenging 1200D dataset that was designed to mimic a natural data distribution, bringing a 63% relative improvement in high-quality samples and 15% in mode detection over its best competitor WGAN-GP.

**Ablation Study:** We run a study on the 2D Ring and Grid data to show the individual effects of each component in our loss. As shown in Table 2, optimizing the determinant $\det L_S$ directly increases the diversity generating the highest quality samples. This works best on the 2D Ring since the true data distribution can be represented by a repulsion model. However, for more complex data such as the 2D Grid, optimizing the determinant fails because it does not well-represent the real manifold structure but aims at repelling the fake samples from each other. Learning the unnormalized structure is prone to outlier structures introduced by the noise in the data and in the learning process. However, when scaling the structure loss by the true-data eigenvalues seems to better disengage the model from noise that exists within the true-data features and only focus on learning the prominent structure.

**Data-Efficiency:** We evaluate the amount of training data needed by each method to reach the same local optima as evaluated by our two metrics on both the 2D Ring and Grid data. Since we are sampling the true-data from a mixture of Gaussians, we can generate an infinite size of training data.

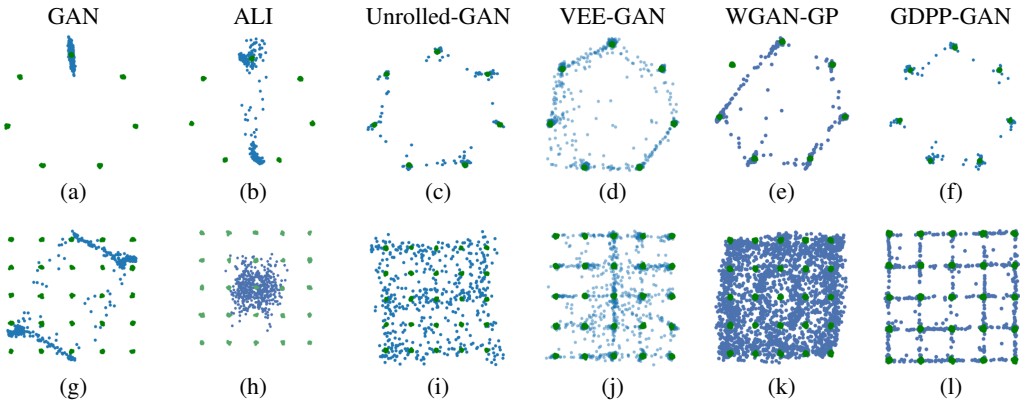

**Figure 2:** Scatter plots of the true data(green dots) and generated data(blue dots) from different GAN methods trained on mixtures of Gaussians arranged in a ring (top) or a grid (bottom).

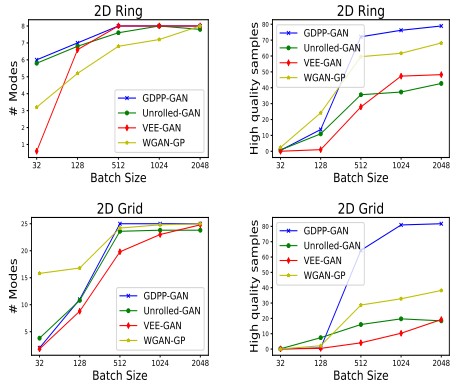 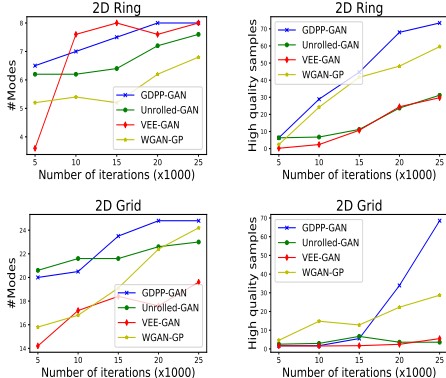

**(a)** Examining the effect of training batch size $B$ given the same number of training iterations.

**(b)** Monitoring convergence at different iterations given the same training data size.

**Figure 3:** Evaluating the models on the 2D Ring and Grid datasets in terms of (a) data-efficiency and (b) time-efficiency. GDPP-GAN tends to converge faster and require the least amount of training data.

Therefore, we can quantify the amount of the training data by using the batch-size while fixing the number of back-propagation steps. In this experiment (Fig. 3a), we run all the methods for the same number of iterations (25,000) and vary the batch size. However, WGAN-GP tends to capture higher quality samples with fewer data. In the case of 2D Grid data, GDPP-GAN performs on par with other methods for small amounts of data, yet it tends to significantly outperform other methods on the quality of generated samples once trained on enough data.

**Time-Efficiency:** Another property of interest is which method converges faster given the same amount of training data. In this experiment, we fix the batch size at 512 and train the models for a variable number of iterations (Fig. 3b). For the 2D Ring, Only VEE-GAN captures a higher number of modes before GDPP-GAN, however, they are of much lower quality than the ones generated by GDPP-GAN. In the 2D Grid data, GDPP-GAN performs on par with unrolled-GAN for the first 5,000 iterations while the others are falling behind. After that, our method significantly outperforms all the methods with respect to both the number of captured modes and the quality of generated samples. We also shows that the GDPP-GAN has an indistinguishable time cost over the DCGAN in Table 6, rendering it the fastest over other baselines.

## 5.2 IMAGE GENERATION EXPERIMENTS

We use the experimental setting of state-of-the-art (Gulrajani et al., 2017) and (Metz et al., 2017) for evaluating models on the Stacked MNIST and CIFAR10. On CelebA, we use the experimental setting of state-of-the-art (Karras et al., 2017). Nonetheless, we investigated the robustness of our method by using a more challenging setting proposed by (Srivastava et al., 2017) and we show its results in Table 5 of Appendix C. In our evaluation, we focus on comparing with state-of-the-art method that adopt a change in the original adversarial loss. Nevertheless, many of them can be deemed orthogonal to our contribution, and can enhance the generation if integrated with our approach. We also show that our method is robust to random initialization in Section C.1. Finally, we show that our loss is generic to any generative model by incorporating it within Variational AutoEncoder (VAE) Kingma & Welling (2013).

|  | 2D Ring | | 2D Grid | |
|---|---|---|---|---|
|  | Modes | % High Quality | Modes | % High Quality |
|  | (Max 8) | Samples | (Max 25) | Samples |
| Exact determinant: $(\det[L_{S_B}])$ | **8** | **82.9** | 12.6 | 21.7 |
| Only diversity magnitude: $(\mathcal{L}_m)$ | **8** | 67.0 | 20.4 | 15.9 |
| Only diversity structure: $(\mathcal{L}_s)$ | **8** | 65.2 | 18.2 | 35.2 |
| GDPP with unnormalized $\mathcal{L}_s$: $(\mathcal{L}_m + \mathcal{L}_s^u)$ | 7.2 | 81.2 | 20.6 | **68.8** |
| Final GDPP-loss: $(\mathcal{L}_m + \mathcal{L}_s)$ | **8** | 71.7 | **24.8** | 68.5 |

**Table 2:** GDPP loss Ablation study on GAN. $\mathcal{L}_s^u$ is the same as $\mathcal{L}_s$ without min-max eigen value normalization

|  | Stacked-MNIST | | CIFAR-10 | |
|---|---|---|---|---|
|  | #Modes (Max 1000) | KL div. | Inception score | IvO |
| DCGAN (Radford et al., 2016) | 427 | 3.163 | 5.26 ± 0.13 | 0.0911 |
| DeLiGAN (Gurumurthy et al., 2017) | 767 | 1.249 | 5.68 ± 0.09 | 0.0896 |
| Unrolled-GAN (Metz et al., 2017) | 817 | 1.430 | 5.43 ± 0.21 | 0.0898 |
| RegGAN (Che et al., 2017) | 955 | 0.925 | 5.91 ± 0.08 | 0.0903 |
| WGAN (Arjovsky et al., 2017) | 961 | 0.140 | 5.44 ± 0.06 | 0.0891 |
| WGAN-GP (Gulrajani et al. (2017)) | 995 | 0.148 | 6.27 ± 0.13 | 0.0891 |
| GDPP-GAN (Ours) | **1000** | **0.135** | **6.58 ± 0.10** | **0.0883** |
| VAE (Kingma & Welling, 2013) | 341 | 2.409 | 1.190 ± 0.02 | 0.543 |
| GDPP-VAE (Ours) | **623** | **1.328** | **1.32 ± 0.03** | **0.203** |

**Table 3:** Performance of various methods on real datasets. Stacked-MNIST is evaluated using the number of captured modes (Mode Collapse) and KL-divergence between the generated class distribution and true class distribution (Quality of generations). CIFAR-10 is evaluated by Inference-via-Optimization (Mode-Collapse) and Inception-Score (Quality of generations).

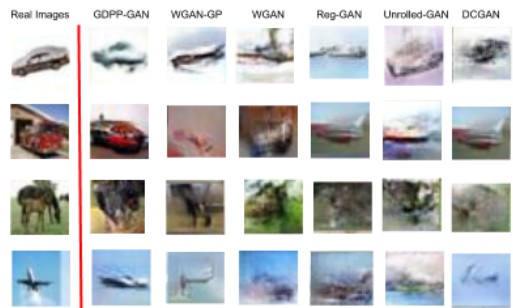

**Figure 4:** Real images and their nearest generations of CIFAR-10. Nearest generations are obtained by optimizing the input noise to minimize the reconstruction error of the generated image.

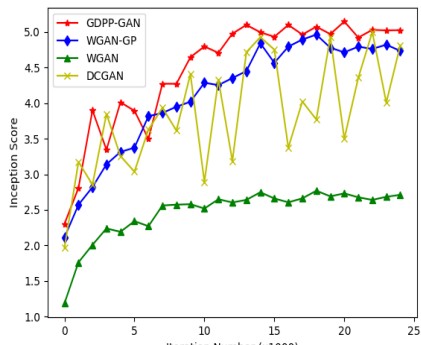

**Figure 5:** Adding GDPP loss to DCGAN stabilizes adversarial training and generates high quality samples earliest on CIFAR-10.

**Stacked-MNIST** A variant of MNIST (LeCun, 1998) designed to increase the number of discrete modes in the data. The data is synthesized by stacking three randomly sampled MNIST digits along the color channel resulting in a 28x28x3 image. In this case, Stacked MNIST has 1000 discrete modes corresponding to the number of possible triplets of digits. Following (Gulrajani et al., 2017), we generate 50,000 images that are later used to train the networks. We train all the models for 15,000 iterations, except for DCGAN and unrolled-GAN that need 30,000 iterations to converge to a reasonable local-optima.

We follow (Srivastava et al., 2017) to evaluate methods on the number of recovered modes and the divergence between the true and fake distributions. We sample 26000 fake images for all the models. We identify the mode of each generated image by using the classifier mentioned in (Che et al., 2017) that is trained on the standard MNIST dataset to classify each channel of the fake sample. The quality of samples is evaluated by computing the KL-divergence between the generated label distribution and the training labels distribution. GDPP-GAN captures all the modes and generates a fake distribution that has the lowest KL-Divergence with the true-data. Moreover, when applied on the VAE, it doubles the number of modes captured (623 vs 341) and cuts the KL-Divergence to half (1.3 vs 2.4).

We note that we run a separate experiment on MNIST in Section C.4 to assess the severity of mode collapse following (Richardson & Weiss, 2018).

**CIFAR-10** We evaluate the methods on CIFAR-10 after training all the models for 100K iterations. Unlike Stacked-MNIST, the modes are intractable in this dataset. To assess the performance on this dataset, we use two metrics: Inception Score for the generation quality and Inference-via-Optimization for diversity. As shown in the Quantitative results on CIFAR and Stacked MNIST (Table 3), GDPP-GAN consistently outperforms all other methods in both mode collapse and developing higher quality samples. When applying the GDPP on the VAE, it reduces the IvO by 63%, however, we note that both the inception-scores are considerably low which is also observed by Shmelkov et al. (2018) when applying the VAE on CIFAR-10.

Inference-via-optimization (Metz et al., 2017), has been used to assess the severity of mode collapse in the generator by providing a metric to compare real images with the nearest generated image. In the case of mode collapse, there are some real images for which this distance is large. We measure this metric by sampling a real image $x$ from the test set of real data. Then we optimize the $\ell_2$ loss between $x$ and generated image $G(z)$ by modifying the noise vector $z$. If a method attains low MSE, then it can be assumed that this method captures more modes than ones that attain a higher MSE. Fig. 4 presents some real images with their nearest optimized generations. Randomly generated sample images can be seen in Appendix D. As demonstrated by (Srivastava et al., 2017), this metric can be fooled by producing blurry images out of the optimization. That is why the inception score is necessary for this evaluation.

Inception score (Salimans et al., 2016) is widely used as a metric for assessing the quality of images. It bases its validity from the premise that every realistic image should be recognizable by a standard architecture(e.g., Inception Network). Ideally, meaning that the score distribution for it must be dominated by one class. We also assess the stability of the training, by calculating the inception score at different stages while training on CIFAR-10 (Fig. 5). Evidently, DCGAN has the least stable training with a high variation. However, by only adding GDPP penalty term to the generator loss, model generates high-quality images the earliest on training with a stable increase.

**CelebA** Finally, to evaluate the performance of our loss on large-scale Adversarial training, we train Progressive-Growing GANs (Karras et al., 2017). We show the effect of embedding our loss in adversarial training by adding it to the WGAN-GP this time instead of DCGAN loss, which is as well orthogonal to our loss. We train the model for 40K iterations corresponding to 4 scales up to $64 \times 64$ results on CelebA dataset (Liu et al., 2018). Unlike CIFAR-10, CelebA dataset does not simulate ImageNet because it only contains faces not natural scenes/objects. Therefore, using a model trained on ImageNet as a basis for evaluation (i.e., Inception Score), will cause inaccurate recognition.

|  | Avg. SWD | Min. SWD |
|---|---|---|
| DCGAN | 0.0906 | 0.0241 |
| WGAN-GP | 0.0186 | 0.0115 |
| GDPP-GAN | **0.0163** | **0.0075** |

**Table 4:** Average and Minimum Sliced Wasserstein Distance over the last 10K iterations.

On the other hand, IvO operates by optimizing the noise vector to match real image. However, large scale datasets requires larger noise vector to cover all the synthetic manifold. This renders the optimization prone to divergence or convergence to poor local optimas; jeopardizing the metric effectiveness. We follow Karras et al. (2017) to evaluate the performance on CelebA using Sliced Wasserstein Distance (SWD) (Peyré et al., 2017). A small Wasserstein distance indicates that the distribution of the patches is similar, which entails that real and fake images appear similar in both appearance and variation at this spatial resolution. Accordingly, SWD metric can evaluate the quality of images as well as the severity of mode-collapse on large-scale datasets such as CelebA. Table 4 shows the average and minimum SWD metric across the last 10K training iterations. We chose this time frame because it shows a saturation in the training loss for all methods. For qualitative examples, refer to Fig. 11 in Appendix D.

## 6 CONCLUSION

In this work, we introduce a novel criterion to train generative networks on capturing a similar diversity to one of the true data by utilizing Determinantal Point Process(DPP). We apply our criterion to Generative Adversarial training and the Variational Autoencoder by learning a kernel via features extracted from the discriminator/encoder. We train the generator on optimizing a loss between the fake and real, eigenvalues and eigenvectors of this kernel to simulate the diversity of the real data. Our GDPP framework accumulates many desirable properties: it does not require any extra trainable parameters, it operates in an unsupervised setting, yet it consistently outperforms state-of-the-art methods on a battery of synthetic data and real image datasets as measure by generation quality and invariance to mode collapse. Furthermore, GDPP-GANs exhibit a stabilized adversarial training and has been shown to be time and data efficient as compared to state-of-the-art approaches. Moreover, the GDPP criterion is architecture and model invariant, allowing it to be embedded with any variants of generative models such as adversarial feature learning or conditional GANs.

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

# SUPPLEMENTARY MATERIAL

## A  NETWORK ARCHITECTURES

The architectures of the generator and discriminator networks employed in our experiments are given in Figure 6.

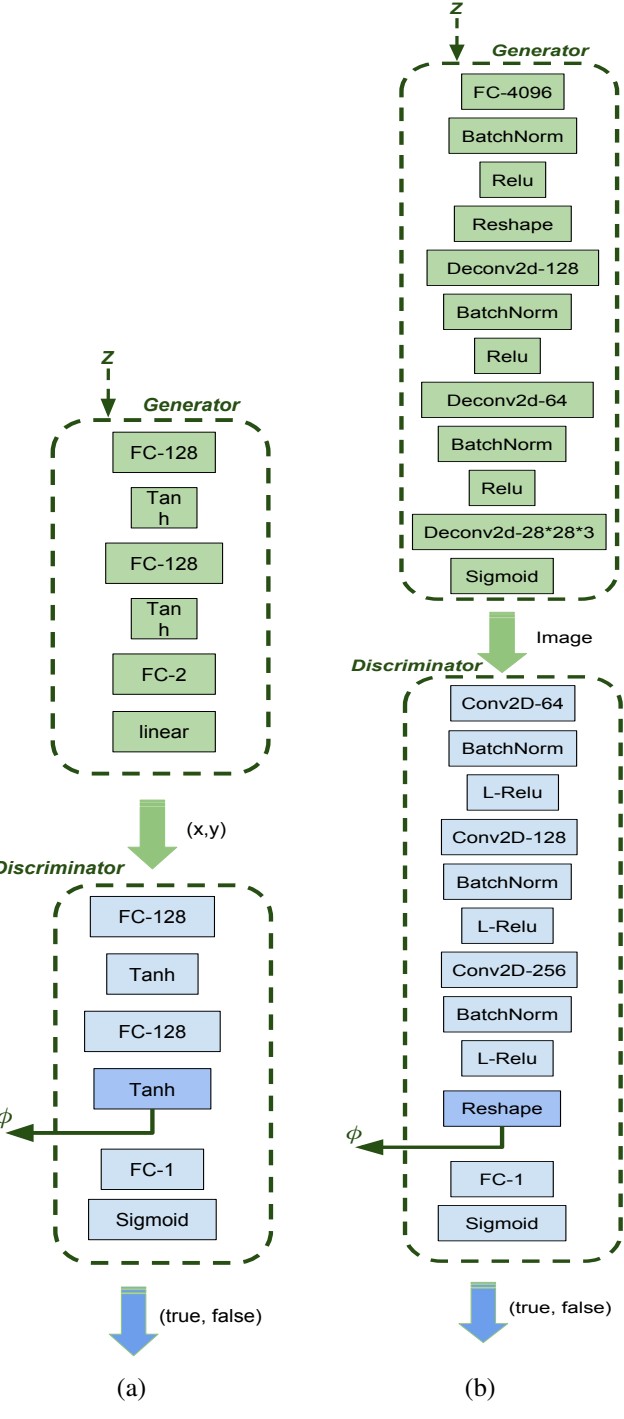

**Figure 6:** (a) Architectures employed in the synthetic experiments. (b) Architectures employed in our image generation experiments.

### A.1 Hyperparameters

In all of our experiments, we use Adam Optimizer with $\beta_1 = 0.5$ and $\epsilon = 1 \times 10^{-8}$. For the synthetic data experiments, we follow the configurations used by (Srivastava et al., 2017) and (Metz et al., 2017). We use $1 \times 10^{-4}$ for the discriminator learning rate, and $1 \times 10^{-3}$ for the generator learning rate. For synthetic data we use a batch size of 512. For Stacked-MNIST and CIFAR-10 we use a batch size of 64. For CIFAR-10, we use a batch size of 16.

For the Stacked MNIST, CIFAR-10 and CelebA datasets, we use $2 \times 10^{-4}$ as the learning rate for both of the generator and the discriminator. To relatively stabilize the training of DCGAN, we follow the protocol in (Gulrajani et al., 2017) to train it by applying a learning rate scheduler. The decay is to happen with a ratio of $1/(\#max - iters)$ at every iteration.

## B Synthetic Data Collections

The first data collection is introduced in (Metz et al., 2017) as a mixture of eight 2D Gaussian distributions arranged in a ring. This distribution is the easiest to mimic since it only requires the generated data to have an equal repulsion from the center of the distribution, even if it is not targeted to the modes. The second and third collections were introduced by (Srivastava et al., 2017). In the second collection, there is a mixture of twenty-five 2D Gaussian distributions arranged in a grid. Unlike the first collection, this one requires a more structured knowledge of the true data modes' locations. The last collection is a mixture of ten 700 dimensional Gaussian distributions embedded in a 1200 dimensional space. This mixture arrangement mimics the higher dimensional manifolds of natural images, and demonstrates the effectiveness of each method on manipulating sparse patterns.

## C Additional Experiments

### C.1 Invariance to Poor Initialization

Since the weights of the generator are being initialized using a random number generator $N(0, 1)$, the result of a generative model may be affected by poor initializations. In Figure 7 we show qualitative examples on 2D Grid data, where we use high standard deviation for the random number generator ($> 100$) as an example of poor initializations. Evidently, GDPP-GAN respects the structure of the true data manifold even with poor initializations. On the other extreme, WGAN-GP tends to map the generated data to a disperse distribution covering all modes but with low quality generations.

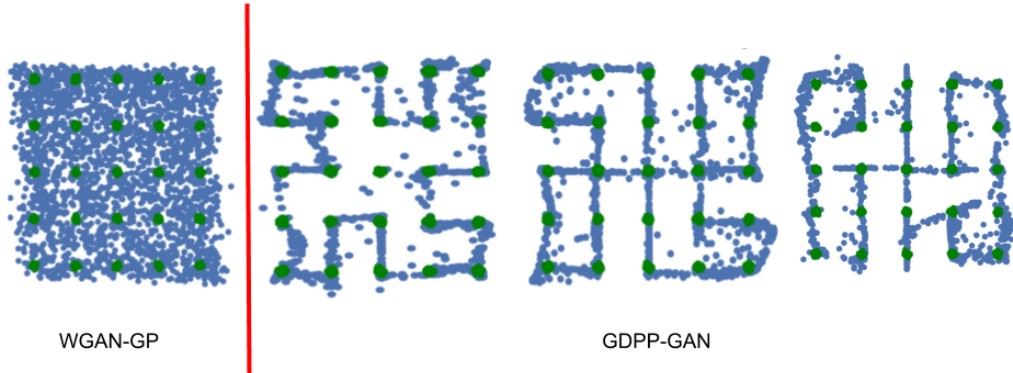

**Figure 7:** The effect of poor initialization on generations: GDPP-GAN models true manifold structure even with poor initializations, while WGAN-GP maps noise to disperse distribution covering the modes with low quality samples.

## C.2 (Srivastava et al., 2017) Experimental Setting on Real Data

To further show the effectiveness of our approach, we examine it under a more challenging experimental setting. The experimental setting of (Srivastava et al., 2017) entails an architecture and hyperparameters that produce relatively poor results as compared with the setting of Table 3. For example, In (Srivastava et al., 2017) setting, DCGAN produces 99 modes, while in our experimental setting, DCGAN produces 427 modes on Stacked MNIST dataset. We note that our main results in Table 3 are computed using the same experimental setting suggested by (Gulrajani et al., 2017) and (Metz et al., 2017) on a more realistic architecture. Our method remains to have a clear advantage when compared to the rest of the methods for both CIFAR-10 and Stacked-MNIST (e.g., covering 90.6% more modes on Stacked-MNIST from 150 to 286 and at a higher quality). We obtain the first four rows from Srivastava et al. (2017).

|  | Stacked-MNIST | | CIFAR-10 |
|---|---|---|---|
|  | #Modes (Max 1000) | KL div. | IvO |
| DCGAN (Radford et al., 2016) | 99 | 3.4 | 0.00844 |
| ALI (Dumoulin et al., 2017) | 16 | 5.4 | 0.0067 |
| Unrolled-GAN (Metz et al., 2017) | 48.7 | 4.32 | 0.013 |
| VEEGAN (Srivastava et al., 2017) | 150 | 2.95 | 0.0068 |
| GDPP-GAN (Ours) | **286** | **2.12** | **0.0051** |

**Table 5:** Performance on real datasets using the challenging experimental setting of (Srivastava et al., 2017). GDPP-GAN remains to outperform all baselines on both Stacked-MNIST and CIFAR-10 for all metrics.

## C.3 Eigendecomposition Running time

Eigendecomposition of an $n \times n$ matrix requires $O(n^3 + n^2 log^2 n log b)$ runtime within a relative error bound of $2^{-b}$ as shown in (Pan & Chen, 1999). In our loss, we perform two eigendecompositions: $L_{S_B}, L_{D_B}$ corresponding to the fake and true DPP kernels respectively. Therefore, the runtime analysis of our loss is $O(n^3)$, where $n$ is the batch size.

Normally the batch size does not exceed 1024 for most training paradigms due to memory constraints. In our experiments, we used 512 for synthetic data and 64 or 16 for real data. Hence, the eigendecomposition does not account for a significant delay in the method.

To further verify this claim, we measured the relative time that eigendecompositions take of each iteration time. We obtained 11.61% for Synthetic data, 9.36% for Stacked-MNIST data and 8.27% for CIFAR-10. We also show the average iteration running time of all baselines in Table 6. We computed the average of 1000 iterations across 5 different runs. Our method is the closest to the standard DCGAN running time, and faster than the rest of baselines by a large margin.

|  | DCGAN | Unrolled-GAN | Reg-GAN | WGAN | WGAN-GP | GDPP-GAN |
|---|---|---|---|---|---|---|
| Avg. Iter. Time (s) | 0.0674 | 0.2467 | 0.1357 | 0.1747 | 0.4331 | **0.0746** |

**Table 6:** Average Iteration time for each of the baseline methods on CIFAR-10. GDPP-GAN obtains the closest time to the default DCGAN.

## C.4 Number of statistically-Different bins (NDB)

(Richardson & Weiss, 2018) proposed to use a new evaluation metric to assess the severity mode collapse severity in a generative model. They based their metric on a simple observation: In two sets of samples that represent the same distribution, number of samples that fall into a given bin should be the same up to a sampling noise. In other words, if we clustered the true-data distribution and fake-data distribution to the same number of clusters/bins, then the number of samples from each distribution in every bin should be similar.

We follow (Richardson & Weiss, 2018) to compute this metric on MNIST (LeCun, 1998) dataset, and compare our method with their results in Table 7. We note that we used their open-source implementation of the metric, and we obtained the first three rows from their paper. We use 20,000 samples from our model and the training data to compute the NDB/K.

| Model | K=100 | K=200 | K=300 |
|---|---|---|---|
| TRAIN | 0.06 | 0.04 | 0.05 |
| MFA (Richardson & Weiss, 2018) | 0.14 | **0.13** | 0.14 |
| DCGAN (Radford et al., 2016) | 0.41 | 0.38 | 0.46 |
| WGAN (Arjovsky et al., 2017) | 0.16 | 0.20 | 0.21 |
| GDPP-GAN | **0.11** | 0.15 | **0.12** |

**Table 7:** NDB/K - numbers of statistically different bins, with significance level of 0.05, divided by the number of bins $K$ (lower is better).

## D ADDITIONAL QUALITATIVE RESULTS

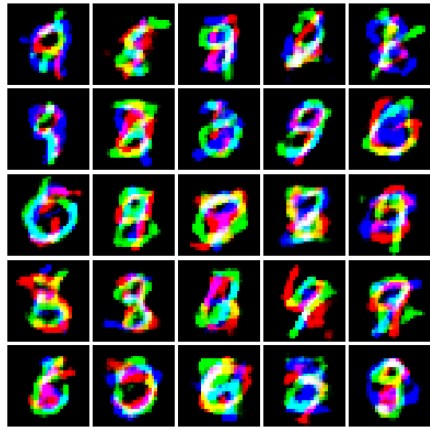

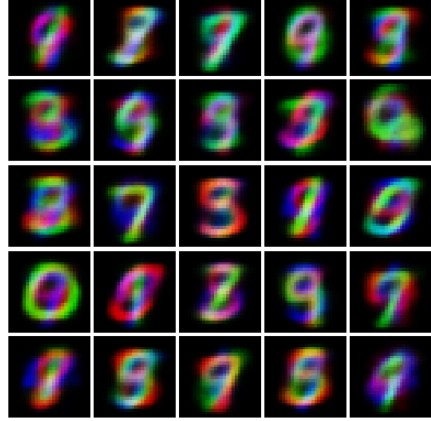

**(a)** GDPP-GAN after 15K iterations.

**(b)** GDPP-VAE after 45K iterations.

**Figure 8:** Random Samples generated on Stacked-MNIST by GDPP-GAN and GDDP-VAE respectively.

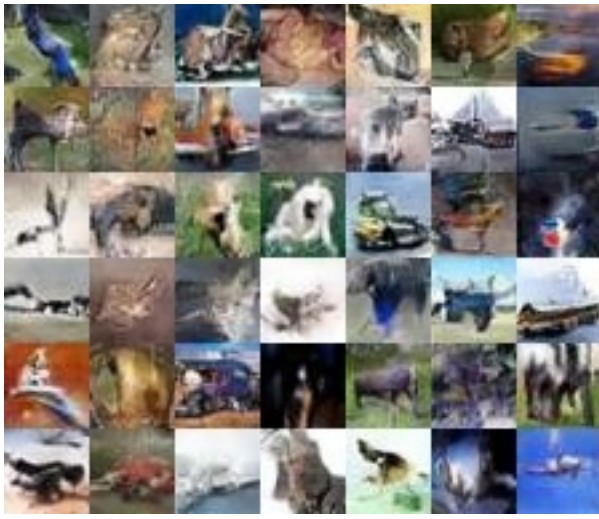

**Figure 9:** Random Samples generated by GDPP-GAN after 100K iterations.

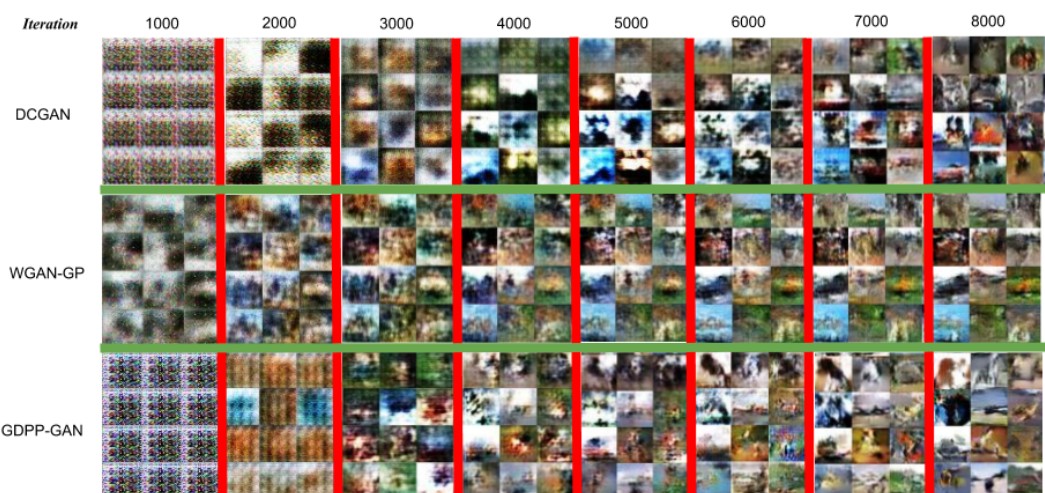

**Figure 10:** Fixed noise qualitative samples progression for different models.

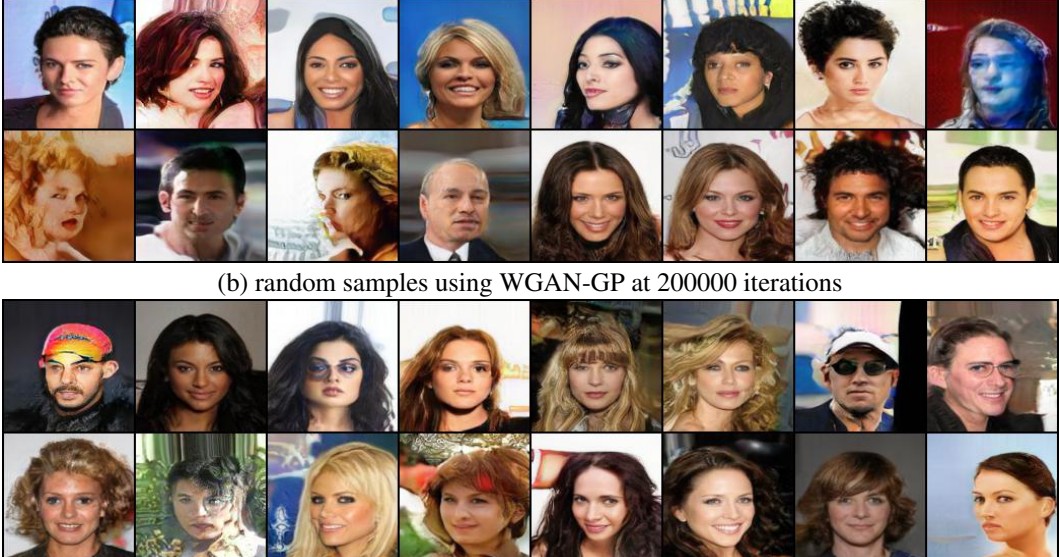

(b) random samples using WGAN-GP at 200000 iterations

(c) random samples using our GDPP-GAN at 200000 iterations

**Figure 11:** Comparing the performance of our loss when compared to DCGAN and WGAN-GP loss, using Progressive-Growing GANs (Karras et al., 2017).

