# OpenReview forum: "Learning Diverse Generations using Determinantal Point Processes"
_ICLR.cc/2019/Conference_

### Official Review · AnonReviewer2 · 2018-11-06
**This paper integrates DPP with GAN and promotes diversity in learning generator distribution**

**Rating:** 5
**Confidence:** 4

**Review:**

The paper proposes to introduce DPP into the vanilla GAN loss and uses it as a way to regularize the generator to produce more diverse outputs, in order to combat the mode-collapse problem. Since the proposed method is added as a simple loss regularizer, the approach does not introduce additional parameters, therefore, less training difficulties. The results on synthetic data seems promising, but there is insufficient evaluation being performed on real and larger dataset where the mode collapse problems are more likely to happen.

- Method
The proposed methods seem sensible. But there are some critical details missing from the current text that prevents me from assessing this paper clearly.

* How are the \lambda and v in Eq.6 calculated? It seems to me that you need to estimate the eigenvalues and eigenvectors at every iteration of your training. I am aware of that many DPP-based models suffer from scalability issues. Could you discuss the potential overhead of this procedure? Also in experiments, you claim "DPP-GAN is cost-free, we observe that the training of GDPP-GAN has an indistinguishable running time than the DCGAN, which are the fastest models to finish training“, which is hard to believe.. could you give more details and analysis on the overhead here?

* In Eq.6 why there are both  "a diversity magnitude loss L_m", and "diversity structure loss L_s". What do they specifically try to capture respectively? Could you give a geometric interpretation on this part?

* what is the batch size used in your experiments on MNIST and CIFAR-10. It seems to me that the effectiveness of GDPP would rely on batch size used as per my understanding you will estimate the DPP kernel using the current batch of samples (generated or real)?

* Despite the fact that GDPP wants to reduce parameters introduced, it is not very intuitive to understand how it would work to use the features outputted by D as the DPP features as well. As D, as the discriminator itself, is trained to distinguish real from fake, while mimicking the eigenvalues/vectors of real data. How would these two goals be reconciled by the same set of parameters?

- Experiments
The results on the synthetic data seem promising, but the results on MNIST and CIFAR-10 are not impressive enough:
* The visual quality of Figure.9 does not look very appealing. I believe many simple variants DCGAN can produce better quality of images
* Why your DCGAN baseline on CIFAR-10 only reports inception score around 5 (with high variance, see Figure.5 and Table 3)? I believe vanilla DCGANs can easily attain an IS at 5.5 to 6, as reported in most recent GAN literature.
* More visual results on CIFAR10 should have been presented in order to demonstrate DPP does generate images with as many classes as existed in CIFAR-10 (which is 10)
* The results could be much more convincing if the authors could show the generation results and evaluation metrics on larger/more real datasets other than CIFAR-10 and MNIST. See GAN literature in 2018 about what dataset to use.


- Presentation
Most parts of this paper are well written. There are few typos and grammatical errors across the text which I believe are easily fixable. There are some missing details that hinder the understanding of some technical parts of the paper. See above for detailed comments.

- Other
Promoting diversity in (deep) generative models isn't a new topic. It would be good if the authors could established connections/differences between this work and this line of relevant  works.

---

> ### Author Response · Authors · 2018-11-06
> **Addressing the comments and describing our additional experiments**
>
> Thanks a lot for your constructive comments. We reply to each of the aforementioned points separately. Additionally, we updated our manuscript with new results of applying GDPP on Variational AutoEncoder and progressive-growing GANs on the CelebA dataset.
>
> [Eigen values and vector computation] Correct, we perform an eigen decomposition to obtain the eigenvalues and eigenvectors.
> In general, there are two factors that contribute to a potential overhead for the eigen decomposition: the batch size and the extracted features dimensionality. The larger the batch size or features dimensionality, the larger the kernel to be decomposed, which constitutes a computational overhead. However, if the data is too complex and large, it would not fit a large batch size in the memory but will need a larger feature representation. And, if the data is too simple and small, can fit larger batch sizes in the memory, and will need a smaller feature representation. Therefore, the computational overhead should be similar across variable types of data.
> In practice, we computed the average ratio of GDPP computation with respect to the total iteration time. We obtained 11.61% for Synthetic data, 9.36% for Stacked-MNIST data and 8.27% for CIFAR-10.
>
> [Interpreting GDPP loss (Eq. 6)] Intuitively, for a positive semidefinite matrix, eigenvectors represent the orientation of distortion within data, and eigenvalues represent the magnitude of distortion of each eigenvector. That's why we have an L2 magnitude loss that favors a matching of the magnitudes of eigenvalues (Magnitude loss), and a Cosine-similarity loss that aims to match the orientations of the eigenvectors (Structure loss).
>
> [Batch size effect] Increasing the batch size is a subject of memory constraints, but theoretically, it should improve the performance up to a limit. In synthetic data, we used a batch size of 512. For stacked-MNIST and CIFAR-10, we used a batch size of 64. In CelebA, we used a batch size of 16, and plan to present results with a larger batch size.
>
> [Quality of the discriminator features] In order for the adversarial network to distinguish real from fake, it learns to extract discriminative features of each data sample. When training with GDPP loss, we use those features to compute the GDPP. Specifically, we compute eigen vectors and values of the DPP kernel constructed using those features. However, we only backpropagate this loss to the generator but not the discriminator.
>
> [Quality on CIFAR10] It is true that DCGANs reach an Inception score greater than 6 in a conditioned setting, whereas we applied GDPP to the more challenging setup of unsupervised adversarial training.  Using the same architecture of WGAN-GP(Gulrajani et al., 2017), with only half the number of training iterations and the standard adversarial training paradigm, our method generates similar qualitative results to the unsupervised WGAN-GP with a higher Inception-Score and a lower Inference-Via-Optimization. We also note that the inception score values greatly depend on the used architecture: using deeper networks would probably improve the performances, but the rank should be the same.
>
> [Diversity in GANs] Unrolled-GAN, VEEGAN, ALI are methods that were explicitly introduced to address mode collapse. We presented these methods in the related work section and compared to them in the experiments section.
>
> [More datasets & VAE] We now present additional results with the state-of-the-art progressive growing architecture on CelebA, demonstrating that our loss scales across deeper networks and more complex datasets. We also show that it performs similarly in Variational Autoencoder, deeming GDPP to be a model and architecture invariant loss.

---

> > ### Author Response · Authors · 2018-11-30
> > **follow-up**
> >
> > May we ask the reviewer, if we were able to address the main concerns that the reviewer had through our rebuttal and revision of the paper? Are there any further issues that the reviewer wants us to address? If so, we would appreciate your feedback and further discussion.

---

### Official Review · AnonReviewer3 · 2018-11-09
**The connection between the propsoed regularizer and the DPP is not precise.**

**Rating:** 5
**Confidence:** 5

**Review:**

For training GANs, the authors propose a regularizer that is inspired by DPP, which encourage diversity. This new regularizer is tested on several benchmark datasets and compared against other mode-collapse mitigating approaches.


In section 2, important references to recent techniques to mitigate mode collapse are missing, e.g.
BourGAN (https://arxiv.org/abs/1805.07674)
PacGAN (https://arxiv.org/abs/1712.04086)
D2GAN (https://arxiv.org/abs/1709.03831)

Also related is evaluation of mode collapse as in
On GANs and GMMs (https://arxiv.org/abs/1805.12462)

The actual loss that is proposed as in (5) and (6), seems far from the motivation that is explained as in Eq (3), using generator as a point process that resembles DPP. This conceptual gap makes the proposed explanation w.r.t DPP unsatisfactory. A more natural approach would be simply add $det(L_{S_B})$ itself as a regularizer. Extensive experimental comparisons with this straightforward regularizer is in order.

It is not immediate if the proposed diversity regularizer $L_g^{DPP}$ in (5) is differentiable in general, as it involves computing the eigen-vectors. Elaborate on the implementation of the gradient update with respect to this new regularizer.


Experiments:

1. The results in Table 3 for stacked-MNIST are very different from VEEGAN paper. Explain why a different setting was used compared to VEEGAN experiments.

2. Similar experiments have been done in Unrolled-GAN paper. Add the experiment from that setting also.

3. In general, split the experiments to two parts: one where the same setting is used as in the literature (e.g. VEEGAN, Unrolled GAN) and the results are compared against those reported in those papers. Another where new settings are studied, and the experiments of the baseline methods are also run by the authors. This is critical to differentiate such cases, as hyper parameters of competing algorithms could have not been tuned as rigorously as the proposed method. This improves the credibility of the experimental results, eventually leading to reproducibility.

---

> ### Author Response · Authors · 2018-11-24
> **Addressing your comments**
>
> We would like to thank you for the insightful comments, and we address them as follows:
>
> [Related Work] Thank you, we added the mentioned references.
>
> [DPP Motivation] Since this is a common comment from reviewers we posted an official reply that clarifies our motivation. Please refer to it for further clarification.
>
> [Differentiability of Our Loss] Since L_S and L_D are symmetric real matrixes, therefore our regularizer is differentiable. Please refer to "On differentiating Eigenvalues and Eigenvectors by Magnus (1985) - Section 3" for a proof of the differentiability of eigendecomposition obtained for symmetric real matrixes. In practice, we used the built-in function "self_adjoint_eig" in Tensorflow implementation and "symeig" operator in PyTorch and both of their implementations are differentiable.
>
> [Experimental Setting] We use the same experimental setting of Unrolled-GAN and WGAN-GP. Both approaches are targetting stabilizing generative training and alleviating mode collapse. We selected this setting because WGAN-GP is the current state-of-the-art stabilization method for adversarial training and is highly cited (734 citations). Also, its implementation is open-source, which guarantees a fair comparison.
>
> We also included the results of applying our method to the VEEGAN experimental setting in Table 5 at Appendix C. Our method remains to outperform all baselines for both experimental settings.
> Finally, we added the evaluation of our method using the NDB/K metric proposed by "On GANs and GMMs" in Table 7 at Appendix C, as suggested.

---

> > ### Author Response · Authors · 2018-11-30
> > **follow-up**
> >
> > May we ask the reviewer, if we were able to address the main concerns that the reviewer had through our rebuttal and revision of the paper? Are there any further issues that the reviewer wants us to address? If so, we would appreciate your feedback and further discussion.

---

### Official Review · AnonReviewer1 · 2018-11-11
**The authors’ motivation from DPP is arguable**

**Rating:** 5
**Confidence:** 4

**Review:**

This paper proposes generative adversarial networks regularized by Determinantal Point Process (DPP) to learn diverse data space. DPP is a probabilistic model that encourages the diversity between the dataset. Authors observe that previous generative models have a mode-collapse problem, and they add generative DPP (GDPP) loss (eq (5)) as a diversity regularizer. Experiments show the GDPP loss is practically helpful to learn under synthetic multi-modal data and real-world image generation.

The paper is well written and easy to comprehend the motivations and main contributions. And the experimental results seem to be interesting. However, there are some arguable issues:

- The main contribution is adding GDPP loss to the original generative models. The authors claim that the GDPP loss (eq (5)) is motivated by the DPP, but I think it does not utilize DPP characteristics at all. The proposed loss is rather close to eigenvalues/vectors matching rather than DPP. It does not seem to be capture DPP properties even assuming the training is perfect. In particular, DPP measures the similarity as the volume of spanned space, while the GDPP loss uses the cosine similarity.

- The GDPP loss is a function of eigenvalues/vectors of kernels, which is generated by internal features of the discriminator. I am curious how to compute the eigenvalues/vectors. Also, the gradient of functions of eigenvalues/vectors is not straightforward as it takes at least a cubic time-complexity with a dimension. It is better to clarify the time complexity for computing the loss and its gradients.

- In addition, if the feature kernel is not a full rank, it is deficient, i.e., some eigenvalues can be zeros. Do you compute the loss all eigenvalues? or compute only some eigenvectors?

- In section 5, the analysis of time-efficiency is not sufficient. Authors report the performance varying the number of iterations. However, since the loss computes eigenvectors/values, the cost per iteration should be larger than other competitors. It is natural to compare the elapsed time or number of FLOPS.

- Although the proposed method shows the best results for the experiments, it is desirable to compare to more diversity encouraging generative models, e.g., DeLiGAN [1]. In addition, I could not recognize the effectiveness of proposals in the experiments of image dataset.

In overall, I think the proposed idea is interesting, but the authors’ motivation from DPP is arguable. In addition, I do not find enough novelty.

Minor issues:
- What is cos(v,w)? Please specify the definition of this.
- Where is Fig. 2k ? Please add the sub-index in Figure 2.

[1] Gurumurthy, Swaminathan, Ravi Kiran Sarvadevabhatla, and R. Venkatesh Babu. “DeLiGAN: Generative Adversarial Networks for Diverse and Limited Data.” CVPR. 2017

---

> ### Author Response · Authors · 2018-11-24
> **Addressing the comments**
>
> Thanks for your constructive thorough comments, we address each in detail.
>
> [DPP Motivation] We responded to this in a separate post.
>
> [Analysis of Eigendecomposition] If an eigenvalue is zero, then it will zero-out its corresponding eigenvectors. This is because eigenvectors are weighted by their corresponding real eigenvalues as illustrated in the second term of Eq. 5.
>
> [Eigendecomposition time-efficiency] The Eigendecomposition of an nxn matrix requires O(n^3+n^2 log^2 nlog b) runtime within a relative error bound of 2^-b as shown in "The Complexity of the Matrix Eigenproblem”, STOC, 1999. In our loss, we perform two eigendecompositions: L_S_B, L_D_B corresponding to the fake and true kernel respectively. Therefore, the runtime analysis of our loss is O(n^3), where n is the batch size.
> Normally the batch size does not exceed 1024 for most training paradigms because of memory constraints. In our case, it is 512 for synthetic data and 64 or 16 for real data. Hence, the eigendecomposition does not account for a significant delay in the method.
> To further verify this claim, we measured the relative time that eigendecompositions take of each iteration time. We obtained 11.61% for Synthetic data, 9.36% for Stacked-MNIST data and 8.27% for CIFAR-10. We also show the average iteration running time of all baselines in Appendix C, Table 5. Our method is the closest to the standard DCGAN running time, and faster than the rest of baselines by a large margin.
>
> [Related Work] We included comparison with DeLiGAN, and our method remains to outperform the rest of the baselines. Regarding comparing with methods that explicitly tackle the mode collapse problem. We note that we are comparing with Unrolled-GAN, VEEGAN, ALI, and RegGAN, and all of them were originally introduced to solve the Mode collapse problem.
>
> [Novelty] To the best of our knowledge, we are the first to introduce modeling data diversity using a Point process kernel that we embed within a generative model. Furthermore, we show the effectiveness of our approach using two common generative models: VAEs and GANs. We assess the performance of our method on a battery of synthetic data as well as small-scale and large-scale real images. We evaluated our method using different metrics and various experimental settings to ensure robustness to hyperparameter effect.
>
> [Minor Issues] Thank you, we addressed the mentioned points. Cos(v, w) is the cosine similarity between eigenvectors of real data and eigenvectors of fake data.

---

> > ### Author Response · Authors · 2018-11-30
> > **follow-up**
> >
> > May we ask the reviewer, if we were able to address the main concerns that the reviewer had through our rebuttal and revision of the paper? Are there any further issues that the reviewer wants us to address? If so, we would appreciate your feedback and further discussion.

---

### Author Response · Authors · 2018-11-24
**DPP Motivation and Additional Experiments**

We thank all the reviewers for their valuable feedback. In this post, we cover questions related to the motivation and additional experiments we performed to address the reviewer’s concerns.  Additionally, we have improved Section 4 and replaced Figure 1 in the paper to better clarify the mentioned points. The rest of this post is organized as follows.

(a) Motivation/Idea
(b) From Motivation to Loss
(c) Additional Experiments

[a: Motivation/Idea]
DPP is an elegant probabilistic model featured with diverse sampling characteristic. Although sampling from DPPs is computationally inefficient, evaluating the probability that a produced sample belonging to a DPP is relatively much faster. We rely on this observation to teach our generator G to generate diverse examples. The generator G produce a sample of size B (batch size) that we can encourage its diversity to improve by backpropagating the DPP diversity metric through the generator parameters. The DPP metric is maximized by producing a batch of orthogonal vectors, this might lead the generator to produce unrealistic images. This is not an issue on the conventional use of DPP in subset selection problems such as video summarization since the selected frames are guaranteed to be realistic. In our generation context, in order to keep the generations on the real image manifold, we match the diversity of the real images to the diversity of the fake images instead by ensuring the closeness of the real eigenvalues to the fake eigenvalues. We also encourage the realism of the structure by matching the real/fake eigenvectors weighted by its corresponding real eigenvalues (its importance), which we found vital in our ablation for the same purpose (see Table 2).

[b: From Motivation to Loss]
As illustrated in Section 3, DPP involves creating a semi-positive definite kernel (L_S) which captures the pairwise similarity between the items of a subset S. The determinant of the kernel L_S was shown to correlate with the diversity of subset S (Eq. 1). Therefore, there is a direct correlation between kernel L_S and the diversity within subset S.
Nonetheless, we are not aiming to merely increase the diversity within subset S (i.e., min det(L_S)). Adding this term to adversarial loss as a regularizer will be equivalent to synthesizing a repulsion model that drives all generated samples apart from each other, where they have the maximum diversity as shown in ablation study (Table 2).
Instead, we are using the kernel L_S to model the diversity within two sets: real data and fake data. Then, we encourage the generator to synthesize fake data that has similar diversity to the diversity of real data. To simplify learning the matrix L_D_B, we choose to learn the major characteristics of the kernel that model its structure: eigenvalues and eigenvectors (Eq. 5).


[c: Additional Experiments]
Additionally, we added the following experiments to show the effectiveness of our approach:
1) [ Reviewer 3] We added an additional ablation with a regularizer (min det(L_S)) to the adversarial loss in Table 2, showing that all the components of our loss are important to achieve the best performance.
2)  [Reviewer 1] We compared with DeLiGAN in Table 3( ~250 modes less than ours, 1.0 less than ours in inception score metric).
3) [Reviewer 1] We computed the average iteration time of all baselines in Table 3, and we report them in Table 5, Appendix C. Evidently, GDPP-GAN has an indistinguishable running time from DCGAN, which are the fastest models to train.
4) [Reviewer 3] We repeated the experiments of Table 3, using the more challenging experimental setting of VEEGAN. We report the results in Table 5, Appendix C. In both settings, our method consistently outperforms other baselines as evaluated on CIFAR-10 and Stacked-MNIST.
5) [Reviewer 3] We evaluated GDPP-GAN using NDB/K evaluation metric in Table 7, Appendix C.
6) [Reviewer 2-CelebA] Demonstrate that our loss scales across deeper networks and more complex datasets, by integrating our GDPP loss within state-of-the-art: Progressive Growing GAN, and applying it on large-scale real images dataset: CelebA. Quantitatively, the addition of our loss shows consistent improvement of the Sliced Wasserstein Distance and qualitatively fewer artifacts in generations; refer to Figure 11 and Table 4.
7) [GDPP on VAE and GAN] Embedded GDPP loss to VAE, and showed that our loss is invariant to the generation approach. In both GAN and VAE, our loss is shown to significantly improve the performance of original generator approach. In fact,  applying our loss on VAE on Stacked-MNIST dataset doubles the number of captured modes (623 vs 341) and cuts the KL-Divergence to half (1.3 vs 2.4).

If you have any further comments or questions, please notify us.
Thank you!!

---

### Author Response · Authors · 2018-12-12
**followup**

We wish all the reviewers happy holidays and we understand that this is a busy time. Motivated by the ICLR spirit to interact more during the review process, we aim to interact more with the reviewers. We worked hard to address the comments and make the paper stronger based on the reviewer's feedback that we really appreciate.  Thank so much for the reviewers and ICLR organizers for the great efforts to make it a great experience both in the process and the venue that gathers many great people.

---

### Meta-Review · Area_Chair1 · 2018-12-12
**Limited novelty**

**Confidence:** 4
**Recommendation:** Reject

**Metareview:**

The paper proposes GAN regularized by Determinantal Point Process to learn diverse data samples.

The reviewers and AC commonly note the critical limitation of novelty of this paper.  The authors pointed out

"To the best of our knowledge, we are the first to introduce modeling data diversity using a Point process kernel that we embed within a generative model. "

AC does not think this is convincing enough to meet the high standard of ICLR.

AC decided the paper might not be ready to publish in the current form.